# Validation of physical activity levels from shank-placed Axivity AX6 accelerometers in older adults

Fatima Gafoor[1]*, Matthew Ruder[2], Dylan Kobsar[1,2]

1 School of Biomedical Engineering, McMaster University, Hamilton, Ontario, Canada, 2 Department of Kinesiology, McMaster University, Hamilton, Ontario, Canada

* gafoorf@mcmaster.ca

**Data Availability Statement:** All relevant data are within the paper and its Supporting Information files.

**Funding:** This research was supported by funding from the Natural Sciences and Engineering

## Abstract

This cross-sectional study aimed to identify and validate cut-points for measuring physical activity using Axivity AX6 accelerometers positioned at the shank in older adults. Free-living physical activity was assessed in 35 adults aged 55 and older, where each participant wore a shank-mounted Axivity and a waist-mounted ActiGraph simultaneously for 72 hours. Optimized cut-points for each participant's Axivity data were determined using an optimization algorithm to align with ActiGraph results. To assess the validity between the physical activity assessments from the optimized Axivity cut-points, a leave-one-out cross-validation was conducted. Bland-Altman plots with 95% limits of agreement, intraclass correlation coefficients (ICC), and mean differences were used for comparing the systems. The results indicated good agreement between the two accelerometers when classifying sedentary behaviour (ICC = 0.85) and light physical activity (ICC = 0.80), and moderate agreement when classifying moderate physical activity (ICC = 0.67) and vigorous physical activity (ICC = 0.70). Upon removal of a significant outlier, the agreement was slightly improved for sedentary behaviour (ICC = 0.86) and light physical activity (ICC = 0.82), but substantially improved for moderate physical activity (ICC = 0.81) and vigorous physical activity (ICC = 0.96). Overall, the study successfully demonstrated the capability of the resultant cut-point model to accurately classify physical activity using Axivity AX6 sensors placed at the shank.

## Introduction

Physical activity (PA) for the aging population has been identified as a protective factor for various musculoskeletal disorders and non-communicable diseases [1]. Even with the well-known benefits of PA, many older adults are representative of the most inactive portion of the population [2]. Thus, an accurate assessment of PA can help to provide insight on the overall health and rehabilitation of the aging population. Traditionally, PA data has been collected using self-reported measures such as questionnaires, activity logs, interviews, and diaries [3]. Common PA self-report tools include the Physical Activity Scale for the Elderly (PASE), the International Physical Activity Questionnaire (IPAQ), and the Short Questionnaire to Assess

Research Council of Canada (Grant Number - RGPIN-2020-06338; www.nserc-crsng.gc.ca; DK) and the Smart Mobility for the Aging Population Collaborative Research and Training Experience program (https://smap.mcmaster.ca/; FG). The funders had no role in the design of the study, the data collection and analysis, the decision to publish, or the preparation of the manuscript.

**Competing interests:** The authors have declared that no competing interests exist.

Health-Enhancing Physical Activity (SQUASH) [4–6]. However, these measures are subject to recall and social desirability biases [7]. Furthermore, many have difficulties distinguishing between different PA intensities in questionnaires [8, 9]. Consequently, these subjective measures lack accuracy, as they often over- or underestimate the true levels of PA [10].

With the advent of wearable accelerometers, researchers can overcome these limitations and obtain objective measures of PA, such as step counts, time spent in different intensity levels, duration of activity, and energy expenditure [11]. These accelerometers can be worn outside of the lab and collect free-living data for multiple days, thereby providing a more comprehensive understanding of an individual's PA levels [7]. Among these devices, Acti-Graph devices are widely used as the gold standard for clinical and research-based PA assessment, with over 20,000 papers published on its use in the past two decades [12].

The quantification of PA levels using these devices typically involves assessing time spent in different PA intensity ranges such as sedentary behaviour (SB), light physical activity (LPA), moderate physical activity (MPA), and vigorous physical activity (VPA). This is achieved by measuring the magnitude of three-dimensional acceleration signals, known as the resultant acceleration. The resultant accelerations are then used to determine activity counts, with higher accelerations resulting in a greater number of measured activity counts per minute [7]. Activity counts are numerical values that reflect the intensity and frequency of movement over a specified time period, usually per minute. Activity that causes the acceleration signal from the sensor to exceed a threshold is essentially 'counted' as activity, whereas those below the threshold are not counted. Higher activity counts indicate more vigorous or frequent movement, whereas lower counts suggest less activity or SB [7]. Examples of SB include sitting, sleeping, and other activities with very little body movement [13]. LPA can be exemplified by activities such as casual walking and light household chores [14], MPA can include activities such as brisk walking, cycling at a moderate pace, or light jogging. Lastly, VPA would include activities such as running and sprinting [15]. Subsequently, cut-points are applied to these activity counts to classify the PA intensity. Cut-points represent thresholds in the activity count spectrum which serve as reference points for categorizing intensity levels (SB, LPA, MPA, and VPA). These cut-points are based on 60-second epoch lengths and they are specific to the wear location of the device [16]. They are determined through validation and calibration studies, where researchers will ask participants to wear accelerometers and simultaneously engage in activities of known intensity, or through comparison with a validated wear location and its cut-points. By comparing the recorded activity counts with the expected intensity levels, researchers can establish cut-points that differentiate between the four PA intensities [17, 18].

The ActiGraph GT9X Link, equipped with an inertial measurement unit (IMU) including accelerometer, gyroscope, and magnetometer sensors, is one device commonly used for collecting PA data. These devices can be worn at the wrist, waist, and ankle but require different cut-point models for each site as the level of accelerations vary between segments [19]. Therefore, deploying a sensor at any new location would require updating and validating a new set of cut-points for that placement site. While wrist, waist, and ankle placements offer sufficient flexibility for assessing PA, recent advancements in sensor capabilities, including improved data quality, higher frequency measurements, and enhanced data storage and battery life, have opened up many new possibilities for free-living PA assessments.

Most notably, wearable sensors are being increasingly used to assess free-living gait. Free-living gait gives us the opportunity to gain insight into the motion of the limbs in an ecologically valid, real-world setting, rather than a conventional gait lab. While conventional gait labs are widely used in gait research, they are highly controlled settings and may be non-representative of an individual's daily, functional gait [20]. Axivity sensors are an example of IMU sensors that have been used in gait research surrounding physical function and disease state [21],

but also possess PA assessment capability given their potential for high-frequency data collections with enhanced data storage and battery life.

While there are a wide variety of placement sites used to acquire free-living gait data, a common placement to assess movement and impact accelerations near the knee is the proximal shank [22–24]. This placement site may offer insights into the progression of musculoskeletal and neurological diseases, such as knee osteoarthritis and Parkinson's through gait assessments [25, 26]. Although this sensor placement can be ideal in assessing free-living gait, there are no methods or cut-points available to determine PA from sensors worn at this wear location. As such, obtaining concurrent PA data while measuring free-living gait would inefficiently require another sensor placed at one of the previously validated sites.

Therefore, the purpose of this study is to develop and validate cut-points for a free-living gait-capable sensor, the Axivity AX6, placed at the proximal shank by comparing it to the gold standard waist-mounted ActiGraph GT9X. It was hypothesized that the agreement of SB and LPA would be higher than the agreement of MPA and VPA. The outcomes compared between the sensors were the percentage of time spent in the different PA intensities: SB, LPA, MPA, and VPA for a total wear time of 72 hours.

## Methods

### Participants

Thirty-five adults over the age of 55 years (18F, age: 71 ± 9 years, height: 167.28 ± 9.79 cm, mass: 76.50 ± 15.10 kg, BMI: 27.00 ± 3.62 kg/m$^2$) were recruited in this cross-sectional validation study. The inclusion criteria included those over the age of 55 years that could ambulate without any walking aids and were not affected by neurological disorders. Participants were recruited from the Physical Activity Centre of Excellence (PACE) at McMaster University, a centre that encourages community exercise and rehabilitation for older adults, from February 12, 2023, to March 23, 2023. The study aimed to recruit healthy older adults with varying daily activity levels to capture a representative spectrum of the aging population, therefore the inclusion criteria were kept relatively general. This is so that the study's cut-points would be applicable to a broader segment of older adults, especially when considering the natural variability in physical abilities that come with aging. All participants provided their written informed consent prior to enrolling in this study. This study was approved by the McMaster Research Ethics Board, Hamilton, ON.

### Procedures

Participants were asked to report their age, sex, height, and mass. Each subject reported if they had received any lower limb injuries in the past year, had been diagnosed with osteoarthritis, and if they had received any lower limb replacement surgeries. Two wearable sensors were used to concurrently collect free-living PA data for 72 consecutive hours. Participants were encouraged to undergo their normal levels of PA that they perform on a daily basis. The first wearable sensor was the gold standard ActiGraph GT9X (ActiGraph LLC, FL, United States), which was worn by each participant above the right iliac crest, a wear location that aligns with the Freedson (1998) cut-points [27] with a waistband and pouch that held the device, as shown in Fig 1. This wear location and anatomical landmark is easy for participants to locate if the waistband or pouch shifts during activities, so they can easily return the pouch in place. Furthermore, the right hip is a placement used by many current researchers when they use the Freedson (1998) cut-points [17, 28]. Participants were encouraged to ensure the pouch sat above their right hip for the full 72 hours. Participants were instructed to only take off the waistband to shower or bath, and if necessary, while sleeping, and wear it immediately upon

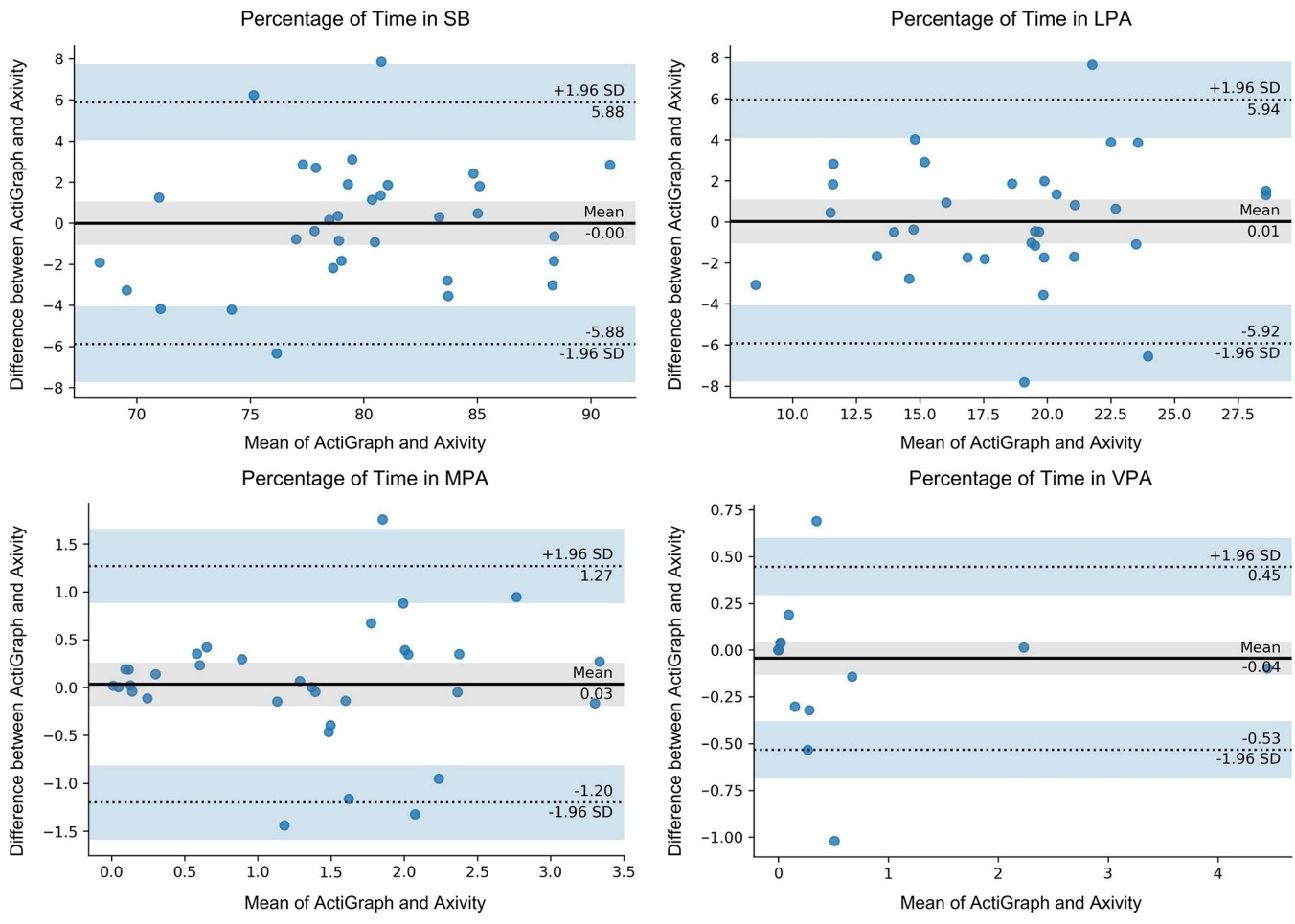

**Fig 1. Wear locations of the ActiGraph and the Axivity for each participant.**

waking and after their shower or bath. The ActiGraph was initialized through the ActiLife v6.13.4 software to start collecting ±8g accelerometer data at 100 Hz at the same time as the Axivity AX6 was initialized to start recording by the Open Movement software (OMGui, version 1.0.0.43; Newcastle University, UK) from the same computer. The 'idle' sleep mode in the ActiLife software was disabled for the ActiGraph sensors.

The sensor to be validated was the Axivity AX6 (Axivity Ltd, Newcastle upon Tyne, United Kingdom) which was placed below the right knee, medial and inferior to the right tibial tuberosity, of each participant using SIMPATCH adhesive patches. The Axivity accelerometer was initialized through the Open Movement software and recorded ±8g accelerometer data at 100Hz, as well as ±1000˚/s gyroscope data that was not analyzed in the current study. Participants were shaved with an electric razor (Philips OneBlade, Amsterdam, Netherlands) before the sensors were placed to ensure the sensors were both comfortable and adhered to the skin well for the entirety of the 72 hours.

## Data analysis

Following the retrieval of the sensors after 72 hours of wear, the data from the ActiGraphs were downloaded and processed through the ActiLife software. Each participants' data was categorized into SB, LPA, MPA, and VPA using the Freedson cut-points and 60-second epochs

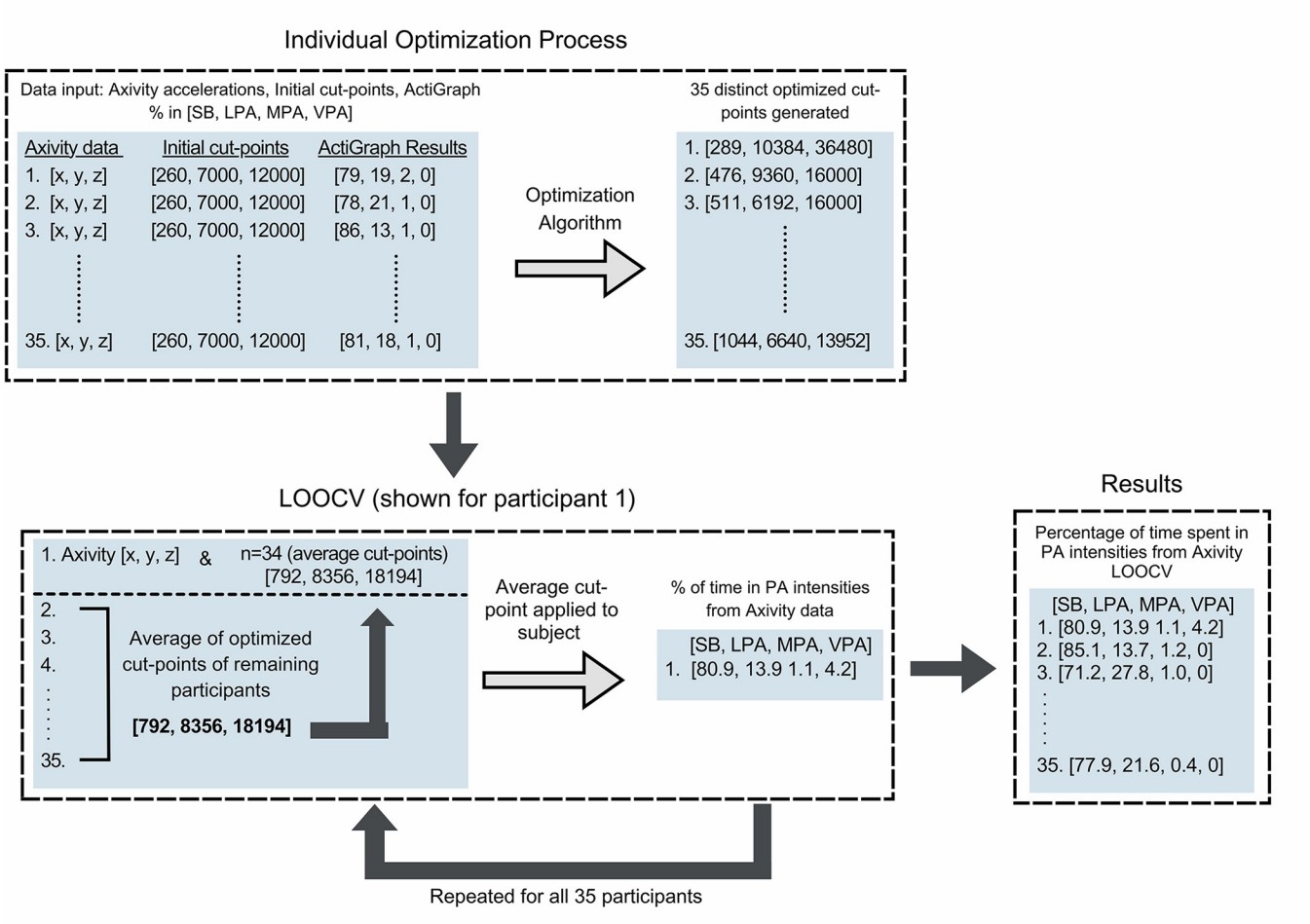

**Fig 2. Flowchart of leave-one-out cross-validation analysis.**

which are available through the ActiLife software [27]. The Freedson cut-points in counts per minute are as follows: SB: 0–99, LPA: 100–1951, MPA:1952–5724, VPA: 5725< [27]. The percentage of time spent in the different intensities was computed by the ActiLife software, based on the activity counts per minute. The time totalled to 72 hours for each participant.

Raw data from the Axivity sensor were downloaded through the OpenMovement software (OMGui) and saved in a binary CWA file. An open-source MATLAB (MathWorks, Natick, MA, USA) script by Felix Liu [29], which uses methods published by Brønd, Andersen, and Arvidsson [30] was used to generate ActiGraph equivalent activity counts per minute from Axivity accelerometers. It should be noted this algorithm also down-samples the Axivity data to 30Hz to make the data comparable to the ActiGraph results. The activity counts for each participant were categorized into PA intensities using optimized cut-points for each participant. However, using the same cut-point values applied to the waist data would result in invalid data given the difference in acceleration signals across these sites [31]. Therefore, to optimize a new set of cut-points for the shank-placed Axivity sensor, an optimization algorithm in MATLAB was created, using the Global Optimization Toolbox and the 'pattern-search' function [32]. Fig 2 outlines the optimization algorithm and resulting cross-validation. This optimization algorithm was given raw Axivity acceleration data, an initial starting set of

cut-points based on pilot testing of free-living PA data collected prior to this study, and the gold-standard results obtained from the ActiGraph (i.e., time spent in SB, LPA, MPA, VPA). This pilot testing consisted of 48-hour collections on four healthy, young adults (22 ± 0.43 years) to provide insight into a rough starting cut-point for the algorithm. The algorithm then sequentially optimized the cut-points for the Axivity data by first evaluating SB, then LPA, MPA, and finally VPA, resulting in three cut-point values to distinguish between SB, LPA, MPA, and VPA. In other words, the optimization algorithm was given the raw data from the Axivity at the shank and tasked to find the cut-points that best approximated and matched the results from the waist-mounted ActiGraph. This optimization occurred at the individual level, resulting in 35 distinct sets of cut-points (i.e., one set for each participant) which produced almost identical results to the ActiGraph output of that participant. For each attempt the algorithm made to generate the cut-points, it produced an error value in the form of a root mean square error, and continued to systematically adjust the cut-point values that were generated iteratively until the lowest possible error value was reached. Once the error value was at a minimum, the corresponding cut-points for that participant were considered their optimized cut-points.

To generate a cross-validated set of cut-points that were blinded from each individual's results, a leave-one-out cross-validation (LOOCV) was applied. While the optimization process outlined above determines the best possible cut-points for each participant, it does this using the known results (i.e., fitting the Axivity results to the ActiGraph results for each individual). In doing so, the resulting data from the optimization process yields the ideal but likely overfit results for each individual. To counter this issue, we conducted a leave-one-participant-out to determine a more generalizable set of cut-points and model accuracy, that is blinded to the known data of each individual. This involved generating a set of cut-points from 34 participants that could then be applied to the data of the left-out subject. In other words, the cut-points from 34 participants were averaged into one set of cut-points that could then be applied to the left-out participant to assess the accuracy of the model. This would then be repeated with a different participant left-out, until all participants had been used as the left-out example. Consequently, the PA data that was generated from this LOOCV method for each participant was solely based on the optimized results from the other 34 subjects, ensuring the cut-points used on each participant were completely blind to the data of that left-out participant. This method of LOOCV is common in machine learning to ensure that the model or cut-points are not overfit to an individual's data. While the process used is not analogous to a machine learning model, given the cut-points were not optimized in a machine learning prediction model, the process is the same, and more importantly, the complete blinding of participant data in model development remains.

### Statistical analysis

The agreement between the ActiGraph and the LOOCV Axivity results for each PA intensity: SB, LPA, MPA, and VPA, were assessed using Bland-Altman plots with 95% limits of agreement (LOA; mean difference of methods ±1.96 SD) [33]. Additionally, intraclass correlation coefficients (ICC; single measure, absolute agreement) with 95% confidence intervals (CIs) were computed for each intensity to measure the reliability of the Axivity. The ICC values were interpreted as poor (ICC < 0.5), moderate (0.5 < ICC <0.75), good (0.75 < ICC< 0.9), and excellent (ICC > 0.9) [34]. The agreement between the ActiGraph and individually optimized Axivity results were also computed, to serve as a comparison of the training data (i.e., individually optimized) versus the generalizable, cross-validated results.

**Table 1. Participant demographics.**

| | |
|---|---|
| Age | 71 ± 9 years |
| Male | 48.6% |
| Height | 167.3 ± 9.8 cm |
| Mass | 76.5 ± 15.1 kg |
| BMI | 27.0 ± 3.6 kg/m$^2$ |
| Previous Lower Limb Surgeries | 5.7% |
| Previous Lower Limb Replacements | 11.4% |
| Diagnosed Osteoarthritis | 31.4% |
| Mean Oxford Knee Score[a] | 37.3 ± 7.1 |

[a]The Oxford Knee Score measures an individual's daily level of function, and how they have been impacted by pain in their knees. Participants only filled out the OKS questionnaire if they indicated they had osteoarthritis in either or both of their knees. Scores range from 0 to 48, with 48 representing maximal functioning knees [35]. A score ranging from 30 to 39 may indicate mild to moderate knee osteoarthritis [36].

## Results

Participant demographics are reported in Table 1. While 35 individuals were recruited for the study, one participant's data was omitted from the analysis due to the early removal of the Axivity sensor. Results from the training model and the LOOCV analysis are shown in Table 2. Individual results from the LOOCV analysis are shown in the Bland-Altman plots in Fig 3. The results from the optimized cut-points training model matched closely with the Acti-Graph results and all intensities had excellent agreement according to the ICC values, as expected given they are individually optimized. When performing the LOOCV analysis, results showed the SB (ICC = 0.85) and LPA (ICC = 0.80) have good reliability, with mean differences of 2.40% and 2.37%, respectively. Whereas the MPA (ICC = 0.67) and VPA (ICC = 0.70) have moderate reliability and mean differences of 0.51% and 0.10% respectively.

After initial data analysis, a significant outlier was identified in the LOOCV analysis. Outliers were defined as data points that deviated significantly from the remaining values of the dataset, using a criterion of falling outside three standard deviations from the mean value. Based on this criterion, one participant was considered an outlier in the LOOCV dataset. In order to ensure the integrity of the results and subsequent cut-point models, a careful decision was made to exclude this outlier from the analysis. By removing this outlier from the dataset, we aimed to achieve a more representative dataset for further analysis. Following the outlier's removal, ICCs for SB (ICC = 0.86), LPA (ICC = 0.82), and MPA (ICC = 0.81) showed good agreement, while the ICC for VPA (ICC = 0.96) showed excellent agreement, as depicted in Table 3. Individual results from the LOOCV analysis with the outlier removed are shown in the Bland-Altman plots in Fig 4.

**Table 2. Results.**

| PA Intensity | Training Model | | | LOOCV Analysis | | |
|---|---|---|---|---|---|---|
| | ICC | Mean Difference (%) | 95% Limits of Agreement | ICC | Mean Difference (%) | 95% Limits of Agreement |
| Sedentary | 0.99 (0.99, 1.0) | 0.01 ± 0.01 | -0.02, 0.02 | 0.85 (0.73, 0.92) | 2.40 ± 1.84 | -6.11, 5.85 |
| Light | 0.99 (0.99, 1.0) | 0.01 ± 0.01 | -0.02, 0.02 | 0.80 (0.65, 0.9) | 2.37 ± 1.97 | -5.93, 6.25 |
| Moderate | 0.98 (0.96, 0.99) | 0.05 ± 0.22 | -0.03, 0.03 | 0.67 (0.44, 0.82) | 0.51 ± 0.70 | -1.96, 1.82 |
| Vigorous | 0.96 (0.93, 0.98) | 0.04 ± 0.22 | -0.02, 0.02 | 0.70 (0.48, 0.84) | 0.10 ± 0.73 | -1.96, 1.30 |

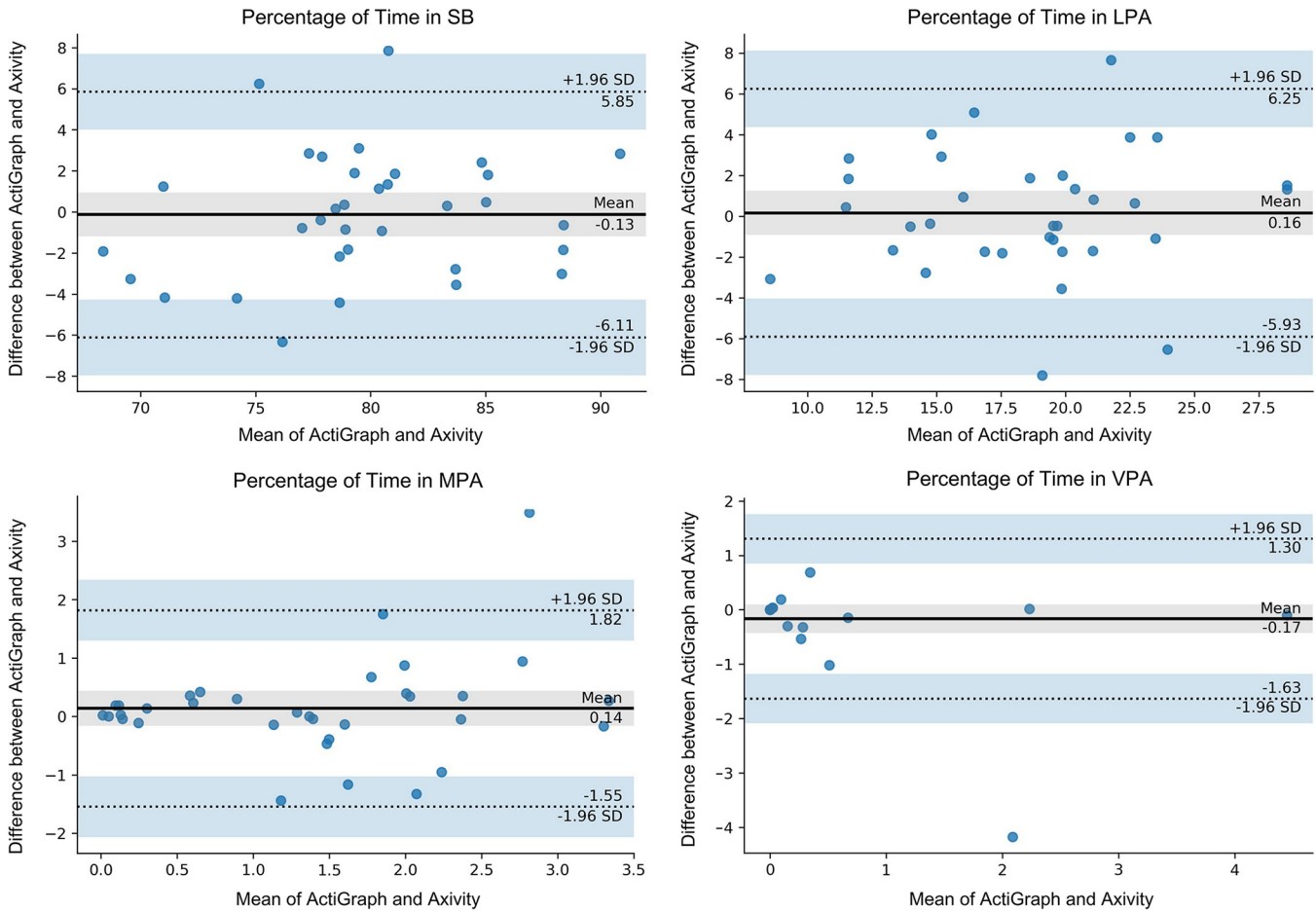

**Fig 3. Bland-Altman plot of the time spent in different PA intensities in minutes, for agreement between the Axivity and the ActiGraph in the LOOCV analysis.**

Based on these results, the resultant cut-points developed for the Axivity accelerometer, when placed below the knee, were computed as the average of the 34 participants and are shown in Table 4. Also shown in Table 4 are the cut-points after the outlier was removed, which is the more representative set of cut-points to use for accelerometers worn at the shank. Additionally, examples of activity corresponding to the different PA intensities have been included in Table 4 for reference. Data outputs from the Axivity AX6 and ActiGraph GT9X accelerometers of participants can be found in S1 Table.

**Table 3. Results with outlier removed.**

| PA Intensity | Training Model | | | LOOCV Analysis | | |
|---|---|---|---|---|---|---|
| | ICC | Mean Difference (%) | 95% Limits of Agreement | ICC | Mean Difference (%) | 95% Limits of Agreement |
| Sedentary | 0.99 (0.99, 1.0) | 0.01 ± 0.01 | -0.03, 0.02 | 0.86 (0.74, 0.93) | 2.34 ± 1.83 | -5.88, 5.88 |
| Light | 0.99 (0.99, 1.0) | 0.01 ± 0.01 | -0.03, 0.03 | 0.82 (0.66, 0.91) | 2.28 ± 1.94 | -5.92, 5.94 |
| Moderate | 0.98 (0.96, 0.99) | 0.01 ± 0.01 | -0.03, 0.03 | 0.81 (0.64, 0.90) | 0.42 ± 0.46 | -1.20, 1.27 |
| Vigorous | 0.96 (0.93, 0.98) | 0.004 ± 0.01 | -0.02, 0.02 | 0.96 (0.92, 0.98) | 0.10 ± 0.23 | -0.53, 0.45 |

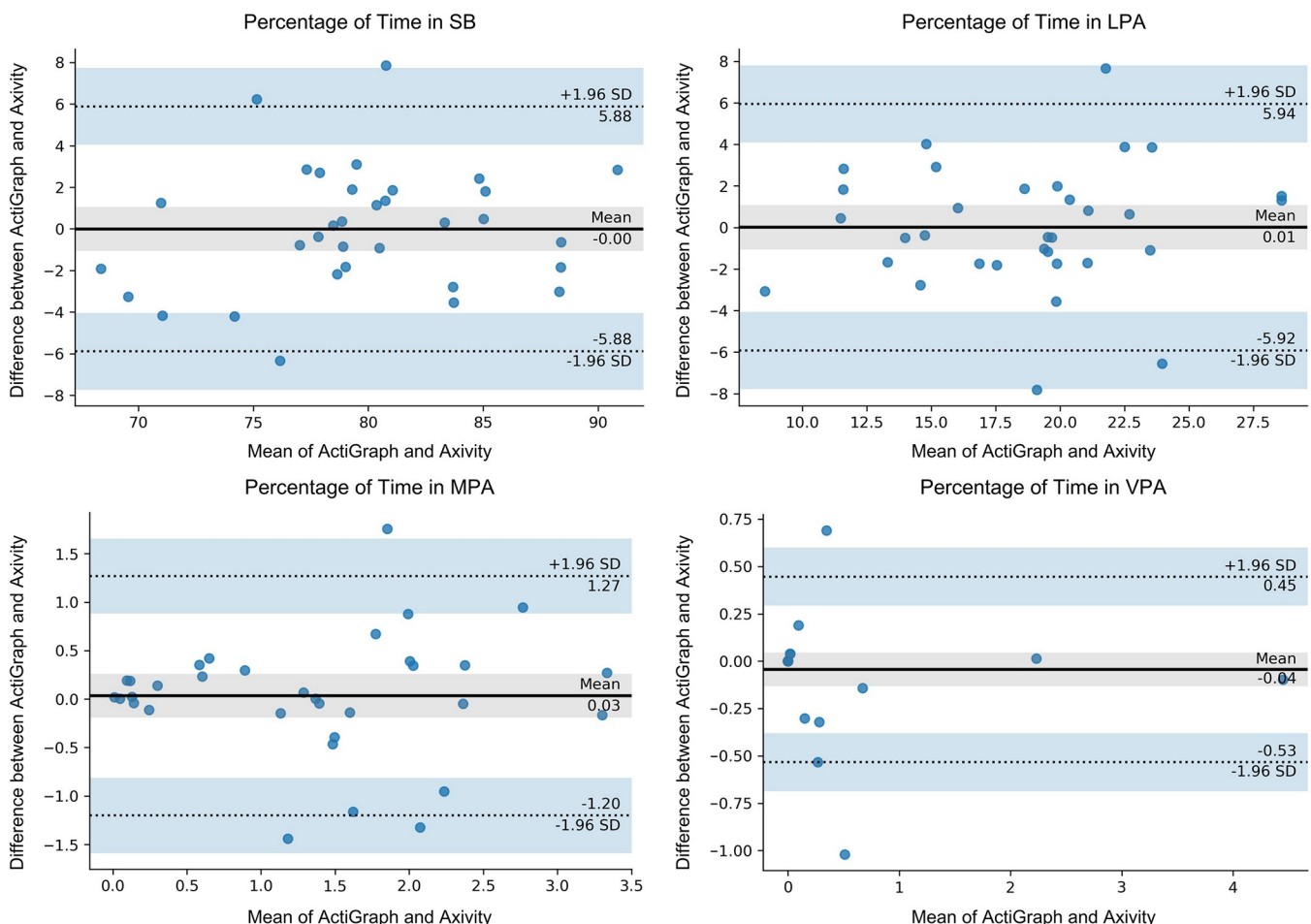

**Fig 4. Bland-Altman plot of the time spent in different PA intensities in minutes, for agreement between the Axivity and the ActiGraph in the LOOCV analysis, with the outlier removed.**

## Discussion

The purpose of this study was to determine and validate cut-points to measure PA with Axivity sensors placed at the shank in older adults. Overall, the Axivity demonstrated good agreement with the gold standard for PA assessment, the ActiGraph, when assessing SB and LPA. When classifying MPA and VPA, the Axivity showed moderate agreement with the ActiGraph. However, with outliers accounted for, MPA and VPA showed good and excellent agreement, respectively. These data demonstrated that accelerometers placed just below the knee can effectively measure an older adult's PA. Overall, the resultant cut-points determined in this

**Table 4. Resultant shank cut-points, with examples of activity shown for each intensity.**

| | | Counts per Minute | |
|---|---|---|---|
| Intensity | Examples of Activity | Complete Dataset | Outlier Removed |
| Sedentary Behaviour (SB) | Sitting, sleeping | 0–777 | 0–792 |
| Light Physical Activity (LPA) | Casual walking, light household chores | 778–8415 | 793–8356 |
| Moderate Physical Activity (MPA) | Brisk walking, cycling at moderate pace, jogging | 8416–18731 | 8357–18437 |
| Vigorous Physical Activity (VPA) | Running, sprinting | 18732 < | 18438 < |

study enable the assessment of PA in older adults at the proximal shank, either independently or in conjunction with research focused on free-living gait data from this site.

Our results supported our original hypothesis which predicted that SB and LPA would have greater agreement than MPA and VPA with the ActiGraph, as the SB and LPA demonstrated ICCs of 0.85 and 0.80, respectively. Whereas the MPA and VPA intensities demonstrated ICCs of 0.67 and 0.70, respectively. This may be because participants, being older adults, did not all perform large amounts of MPA and VPA. On average, participants spent 79.70%, 18.60%, 1.46%, and 0.24% of their time in SB, LPA, MPA, and VPA, respectively. The average percentage of time spent in VPA for participants that completed some form of VPA was 1.04%, with 26 participants having no VPA. This indicates reduced MPA and VPA data collected, which could have affected the average of the resultant cut-points to identify MPA and VPA. These results are consistent with previous studies that assessed PA between ActiGraph and other devices, including the Axivity, which also found the greatest agreement when classifying SB and LPA over MPA, and VPA, such as the study done by Rowlands et al. [37], due to the relatively small amount of time spent in moderate and vigorous PA levels [38].

With one outlier removed, these ICCs improved to 0.86, 0.82, 0.81, and 0.96 for SB, LPA, MPA, and VPA, respectively. Upon evaluation, the recorded Axivity accelerations and subsequently derived activity counts of the outlier were substantially higher in magnitude compared to those of the remaining participants, despite the ActiGraph data of the outlier aligning with that of the other participants. Consequently, the outlier's optimized cut-points were significantly greater than the average cut-points of all subjects. Thus, when conducting the LOOCV including the outlier's optimized cut-points, the model overestimated the time spent in MPA and VPA. Specifically, the resultant cut-points including the outlier are higher for MPA and VPA intensities when compared to the cut-points with the outlier removed. Higher recorded or erroneous accelerations from one subject's Axivity can be due to many potential reasons, which can include sensor movement as a result of being loosely attached and the introduction of artifacts and noise into the accelerometer readings. Nevertheless, we cannot pin-point the exact cause as to why one subject's Axivity sensor collected greater accelerations than others' Axivity sensors. The optimized cut-points of the outlier affected all PA intensities, however, it had a limited impact on SB and LPA data, which further highlights the quality of these outcomes in older adults compared to MPA and VPA. Despite the outlier's removal improving the results to excellent validity, we still caution the use of VPA data in older adults moving forward, given the limited amount of data in these intensity levels.

With respect to the cut-points used for MPA and VPA, they are similar to cut-points found in a study done by Rhudy et. al. [16] which identified cut-points for the ActiGraph GT9X at varying wear locations proximal to the shank, such as the ankle, foot, along with the wrist and hip to measure moderate and vigorous PA, which are shown in Table 5, along with the shank cut-points from the current study. These wear locations offer some opportunity for comparing cut-points at proximal locations, such as the ankle and foot. Theoretically, the cut-points for the ankle and foot should be greater, as the impact accelerations experienced would be greater

**Table 5. Cut-points for MPA and VPA at foot, ankle, and hip wear locations from Rhudy et al. [16] with the current study's cut-points for the shank.**

| Wear Location | MPA cut-point | VPA cut-point |
|---|---|---|
| Foot [16] | 20575 | 22629 |
| Ankle [16] | 14767 | 17818 |
| Shank [current study] | 8357 | 18438 |
| Hip [16] | 4978 | 6227 |

than those at the shank. It is also expected that the cut-points for the shank for both MPA and VPA are greater than those of the hip. This is mostly reflected in the results from Rhudy et al. and the current study, as the cut-points for MPA and VPA measured at the ankle are 14767 and 17818 counts per minute, respectively, and the cut-points for MPA and VPA measured at the foot are 20575 and 22629 counts per minute, respectively. For the current study, the cut-points for MPA and VPA measured at the shank are 8357 and 18438. As expected, these shank cut-points are greater than the cut-points for the hip for MPA and VPA, which are 4978 and 6227 counts per minute. Furthermore, the cut-point for MPA at the shank is less than the cut-points at the ankle and foot, as predicted. While the cut-point for VPA at the shank is less than the cut-point for the foot, it is greater than the cut-point at the ankle. This could be since the cut-points for the foot and ankle in the aforementioned study were identified through a young, healthy population in a controlled setting, whereas the current study's population consisted solely of older adults in free-living conditions. Therefore, in a free-living setting, older adults are less likely to perform VPA, as demonstrated by the collected ActiGraph data, so the threshold for them to reach VPA would be higher than that of a younger adult to perform VPA. This suggests that what is considered vigorous for older adults, based on their activity patterns, may require a higher level of intensity when compared to younger adults.

This study has some limitations to be discussed. Firstly, participants removing the Acti-Graph sensor when showering may have influenced the results, as the Axivity was worn for the full 72 hours. This likely would have minimal influence over the results with both accelerometers likely resulting in a SB classification, as showers are typically short in duration and most people won't move a significant amount while showering. However, if participants forgot to wear or delayed the placement of the ActiGraph after their shower, this could adversely affect agreement between two sensors.

Secondly, it is possible that individuals may have performed movement with their lower limbs, without moving their trunk, such as leg raises, which could have recorded more accelerations in the Axivity than the ActiGraph, if the hip-worn accelerometer was not experiencing movement. This would have impacted the final cut-points that were developed. Nevertheless, the current findings present a first step at classifying PA from shank-placed accelerometers and identifying cut-points for this wear location, which can be widely used in gait research in the elderly.

## Conclusion

These results displayed a good agreement between the gold standard ActiGraph sensors and the Axivity sensors to classify PA and SB in older adults when worn at the shank. The resultant cut-points can be used for accelerometers, including the Axivity and ActiGraph, worn specifically just below the knee at the shank recorded at a range of ±8g. These cut-points were developed on an older adult population and likely should only be primarily used on older adults. The introduction of shank cut-points is a novel contribution that will aid in simplifying concurrent PA assessments that occur in older adults also completing gait assessments.

## Supporting information

**S1 Table. Participant Axivity and ActiGraph data.** Axivity Data is missing from Participant 21 due to early removal of the Axivity AX6 sensor.
(DOCX)

## Acknowledgments

I would like to thank the Physical Activity Centre of Excellence at McMaster University for supporting this study, as well as all the participants for their time and efforts, that made this study possible.

## Author Contributions

**Investigation:** Fatima Gafoor.

**Supervision:** Matthew Ruder, Dylan Kobsar.

**Writing – original draft:** Fatima Gafoor.

**Writing – review & editing:** Matthew Ruder, Dylan Kobsar.

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
