## [Decision Letter · Decision Letter 0]

25 Oct 2023

PONE-D-23-25684Validation of physical activity levels from shank-placed Axivity AX6 accelerometersPLOS ONE

Dear Dr. Gafoor,

Thank you for submitting your manuscript to PLOS ONE. After careful consideration, we feel that it has merit but does not fully meet PLOS ONE’s publication criteria as it currently stands. Therefore, we invite you to submit a revised version of the manuscript that addresses the points raised during the review process.

Authors presented an interesting study which value was recognized by both Reviewers. However, the Experts recognized some major points that need to be carefully addressed before publishing the manuscript. In particular, it was highlighted a lack in the  methodological procedure that may alter the the validity of the results and conclusion. Since Authors performed a validation in their study, it is very important that the methodology is clearly stated and the analysis properly carried. For this reason I suggest major revision of the work. 

We look forward to receiving your revised manuscript.

Kind regards,

Andrea Tigrini, Ph.D.

Academic Editor

PLOS ONE

Journal Requirements:

Did you know that depositing data in a repository is associated with up to a 25% citation advantage (https://doi.org/10.1371/journal.pone.0230416)? If you’ve not already done so, consider depositing your raw data in a repository to ensure your work is read, appreciated and cited by the largest possible audience. You’ll also earn an Accessible Data icon on your published paper if you deposit your data in any participating repository (https://plos.org/open-science/open-data/#accessible-data).

**Additional Editor Comments:**

Authors presented an interesting study which value was recognized by both Reviewers. However, the Experts recognized some major points that need to be carefully addressed before publishing the manuscript. In particular, it was highlighted a lack in the methodological procedure that may alter the the validity of the results and conclusion. Since Authors performed a validation in their study it is very important that the methodology is clearly stated and the analysis properly carried. For this reason I suggest major revision of the work.

Reviewers' comments:

Reviewer's Responses to Questions

**Comments to the Author**

1. Is the manuscript technically sound, and do the data support the conclusions?

Reviewer #1: Partly

Reviewer #2: Yes

2. Has the statistical analysis been performed appropriately and rigorously? 

Reviewer #1: Yes

Reviewer #2: Yes

3. Have the authors made all data underlying the findings in their manuscript fully available?

Reviewer #1: Yes

Reviewer #2: No

4. Is the manuscript presented in an intelligible fashion and written in standard English?

Reviewer #1: Yes

Reviewer #2: Yes

5. Review Comments to the Author

Reviewer #1: In the presented work, Gaboor and colleagues tried to obtain reliable results placing accelerometers at the shank in older adults to measure physical activity. In particular, they wanted to identify and validate specific cut-points to recognize four different levels of exercise intensity. The work is well-written and fluid in the reading. However, there are some parts that need more details. More specifically, the method section lacks of important information in my opinion and some issues must be addressed. Comments are reported below.

- You talked a lot about PA, but which kind of PA were the subjects asked to perform?

- Line 55: "objective measures of PA" such as? I would suggest to mention some of them.

- Line 64: "activity counts", what are they? How are they identified? Based on which criterion?

- Line 66: it is stated that cut-points are used to classify the intensity, but firstly it would be appreciated if it is explained what these cut-points are and then how they are identified (trial and error or mathematically or what so ever). It is a key point for good comprehension of the work.

- Line 111: in this paragraph it is reported the number of used sensors and their location, but a figure should be included to better understand the considered experimental protocol.

- Line 124-131: ActiGraph and Axivity are two different types of sensors that were used, the first one as the gold standard and the second one to be validated. The ActiLife software was used to acquire simultaneously the two sensors, so was the signal to start the recording given directly by the software to both type of sensors? Moreover, nothing is said about the preprocessing of the data, such as filtering, do this mean that no pre-processing has been done and data has been used in their raw form or something different? I suggest to report the preprocessing part if done, if not performed to specify it and justify why.

- Line 140-144: to make a fairer comparison you should verify that the results given by this MATLAB script are the same of those given by the ActiLife software using the same data, i.e. those of the ActiGraph (if it has been already done, please specify and add reference if necessary).

- Line 148: you stated that an optimization algorithm was created and a short description follows; you should go more in depth in this part, it is not enough, a more accurate description of the optimization algorithm should be provided, which is the reasoning behind and the chosen criteria.

- Line 151: are the data collected prior to this study referred to the same type of PA? In my opinion, they should be the same type of PA to be included as starting point.

- Line 249: "the time spent in..." the sentence is not concluded, please conclude it.

Regarding the participants, it is not clear if the participants are all healthy or all pathological or a mix of them. As it is written, it seems that it is not a homogeneous set of subjects, in my opinion mixing data from healthy and pathological subjects (moreover different among them in the pathology) is not a fair, since this could influence the results. Could you specify better the healthy status of the subjects?

About the location of the sensor that you wanted to validate, it is clear that it is an ideal position for free-gait analysis but it is neither a comfortable position for the subject nor a location that could be included for instance in the clothes, so why should be used?

Reviewer #2: The paper analyzed shank-mounted Axivity data to classify the physical activity levels on older adults during recordings lasting 72 hours. Congratulations to the authors, the study is interesting. In general, the paper has a high-quality English language and the results are clearly presented, but further analyses and comments should be added and discussed.

See the attached file with the bullet points.

6. PLOS authors have the option to publish the peer review history of their article (what does this mean?). If published, this will include your full peer review and any attached files.

Reviewer #1: No

Reviewer #2: No

---

## [Author Response · Author response to Decision Letter 0]

31 Dec 2023

Hello PLOS ONE Journal Staff, 

Please view my revised manuscript with the changes made in accordance to the suggestions and comments made by my reviewers. The 'response to reviewers' document also has each concern and question addressed. This document has been copy and pasted to also contain the responses below. 

Thank you, 

Fatima 

Reviewer 1 Comments:

Thank you for your thoughtful and thorough comments in reviewing our manuscript. We have addressed all your concerns, and we feel the manuscript is much improved. The notes below highlight these changes and hope they make it easy to follow these revisions. 

1. You talked a lot about PA, but which kind of PA were the subjects asked to perform?

Response: Thank you for highlighting this point of unclarity. They were not asked to perform specific amounts of any intensity of physical activity. To your point, while some researchers may instruct participants to perform varying levels of PA, such as Rhudy M.B., et al. 2020, we chose to study free-living PA, similar to a study done by Dutta A. et. al., 2018. This has also been clarified on lines 143-144. Participants were asked to perform their free-living PA, meaning how much activity they undergo on a routine basis.

- Dutta, A., Ma, O., Toledo, M., Pregonero, A. F., Ainsworth, B. E., Buman, M. P., & Bliss, D. W. (2018). Identifying Free-Living Physical Activities Using Lab-Based Models with Wearable Accelerometers. Sensors (Basel, Switzerland), 18(11), 3893. https://doi.org/10.3390/s18113893

2. Line 55: "objective measures of PA" such as? I would suggest to mention some of them.

Response: Excellent point, this has now been updated to include objective measures of PA on lines 55-56. The objective measures listed include step counts, time spent in different intensity levels, duration of activity, and energy expenditure (Silfee V.J. et. al., 2018)

Silfee VJ, Haughton CF, Jake-Schoffman DE, Lopez-Cepero A, May CN, Sreedhara M, et al. Objective measurement of physical activity outcomes in lifestyle interventions among adults: A systematic review. Prev Med Rep. 2018;11: 74–80. doi:10.1016/j.pmedr.2018.05.003

3. Line 64: "activity counts", what are they? How are they identified? Based on which criterion?

Response: Thank you for identifying this weakness in our previous draft. Activity counts are numerical values which reflect the intensity and frequency of movement over a specified time period, usually per minute. Activity that causes the acceleration signal from the sensor to exceed a threshold is essentially ‘counted’ as activity, whereas those below the threshold are not counted. Typically, higher activity counts means moderate and vigorous physical activity is being performed, whereas lower activity counts constitute sedentary and light physical activity. In terms of which criterion they are identified, this is based on thresholds and cut-points which are used to classify them as the different types of PA and connects well to the purpose of the study, of aiming to determine these cut-points. We have added additional text that provide additional context on the topic of activity counts on lines 66-72. 

4. Line 66: it is stated that cut-points are used to classify the intensity, but firstly it would be appreciated if it is explained what these cut-points are and then how they are identified (trial and error or mathematically or what so ever). It is a key point for good comprehension of the work.

Response: Cut-points are the thresholds in the activity count spectrum which serve as reference points to categorize the four PA intensity levels (SB, LPA, MPA, and, VPA). These cut-points are based on 60-second epoch lengths and they are specific to the wear location of the device. They are determined through validation and calibration studies, where researchers often ask participants to engage in activities of known intensity. By comparing the recorded activity counts with expected intensity levels, researchers can then establish cut-points that differentiate between the four PA levels. Based on your feedback, we have also further explained the topic of cut-points on lines 76-84. 

5. Line 111: in this paragraph it is reported the number of used sensors and their location, but a figure should be included to better understand the considered experimental protocol.

Response: Yes, thank you for the idea. A figure has now been added to the manuscript. This figure is shown below, with the caption located at line 176 in the text. 

Fig 1. Wear locations of the ActiGraph and the Axivity for each participant 

6. Line 124-131: ActiGraph and Axivity are two different types of sensors that were used, the first one as the gold standard and the second one to be validated. The ActiLife software was used to acquire simultaneously the two sensors, so was the signal to start the recording given directly by the software to both type of sensors? 

Thank you for highlighting this point of confusion. The ActiGraph was initialized through the ActiLife software but was time-synchronized with the computer. Based on this, we configured the settings of the ActiGraph to start recording at a set time. The Axivity was initialized through the OpenMovement GUI, which was also time-synchronized with the same computer. It was also configured by us to start recording at the same time that was chosen for the ActiGraph. This was so that we would record the same activities performed by the participant between the two sensors. The two devices themselves were not synchronized to each other. This is now clarified on lines 154-157. 

7. Moreover, nothing is said about the preprocessing of the data, such as filtering, do this mean that no pre-processing has been done and data has been used in their raw form or something different? I suggest to report the preprocessing part if done, if not performed to specify it and justify why.

You are correct in assuming that raw data was used for this study. This is noted on line 185 which states “Raw data from the Axivity sensor were downloaded through the OpenMovement...”. For the ActiGraph data, this is post-processed using the ActiLife software, which has limited information available as to how it is processed, due to proprietary reasons. For the Axivity data, pre-processing such as filtering was also not done, but was completed at the post-processing stage to ensure optimal signal quality, noise reduction, and alignment with the study’s analytical objectives. Overall, given our limited knowledge of the ActiGraph's functions, our intention is to preserve as much of the original signal as we can.

8. Line 140-144: to make a fairer comparison you should verify that the results given by this MATLAB script are the same of those given by the ActiLife software using the same data, i.e. those of the ActiGraph (if it has been already done, please specify and add reference if necessary).

Thank you for raising this important point. It must be noted that this script was chosen carefully because it follows the published methods to replicate ActiGraph equivalent activity counts (Brønd JC, 2017). This study found a high degree of agreement in the PA intensity classification as the ActiGraph and associated ActiLife software, when using raw accelerometer data. So far, this has also been cited by 51 other articles in the PA field. While other measures (e.g., step count, step length) have not been validated, this particular activity count script has been validated. Also, as elaborated on in the discussion from lines 331 to 352, we compared our results of activity counts and cut points from other wear locations on the lower limb. This analysis suggested that our estimates of activity counts are in line with previous studies. 

9. Line 148: you stated that an optimization algorithm was created and a short description follows; you should go more in depth in this part, it is not enough, a more accurate description of the optimization algorithm should be provided, which is the reasoning behind and the chosen criteria.

Thank you for highlighting this point of confusion. This has now been elaborated on from lines 207 to 214. 

10. Line 151: are the data collected prior to this study referred to the same type of PA? In my opinion, they should be the same type of PA to be included as starting point.

It is now clarified that this is free-living PA on line 200. These collections were shorter in length, in which data was only recorded for 48 hours compared to the 72 in the study, and had fewer collections (n=4) than those listed in this study, to give us a starting point for our algorithm. These details have been added in the manuscript on lines 201 to 203. 

11. Line 249: "the time spent in..." the sentence is not concluded, please conclude it.

Thank you, this has now been done. 

12. Regarding the participants, it is not clear if the participants are all healthy or all pathological or a mix of them. As it is written, it seems that it is not a homogeneous set of subjects, in my opinion mixing data from healthy and pathological subjects (moreover different among them in the pathology) is not a fair, since this could influence the results. Could you specify better the healthy status of the subjects?

Thank you for commenting on this. Essentially, they were all older adults that did not reside in long-term care homes and did not require any walking aids or had any major neurological conditions that would affect their walking. We also included participants that did have diagnosed osteoarthritis, some which had osteoarthritis of the knee. Based on their mean Oxford Knee Score shown in Table 1, which was 37.3, it is on the closer end to being fully functional and only indicates mild to moderate knee osteoarthritis, meaning that their knee osteoarthritis had minimal impact on their ability to perform physical activity. The reason why we included adults with knee osteoarthritis or previous lower limb surgeries and replacements was because we wanted our data to be as generalizable to the older adult population as possible. This has also now been clarified in the participants section in lines 126, and 129-133. 

13. About the location of the sensor that you wanted to validate, it is clear that it is an ideal position for free-gait analysis but it is neither a comfortable position for the subject nor a location that could be included for instance in the clothes, so why should be used?

Response: The selected sensor location, while optimal for capturing shank accelerations for gait analysis, does have similar potential challenges related to subject comfort and practical wearability that commercial devices, such as smart watches have. This is especially true because it may not be easily integrated into everyday clothing. Despite these considerations, the shank remains a widely researched location to provide valuable insights that can be extended to classify the health of individuals impacted by injuries and diseases affecting the knee, such as knee osteoarthritis patients. Therefore, the purpose of this wear location would be to provide additional insights, such as PA data, to researchers that are placing sensors at this location. Being able to obtain PA data from a single sensor like this would reduce the need to have the patient wear a second sensor, which would imrpvoe the overall patient experience, and likely compliance. Studies done by Li. Q. et al., 2010, Celik Y. et al., 2021, which are cited in the paper, as well as some studies done by Kobsar D., 2020 which are listed below, support the shank as a common wear location in gait analysis. Furthermore, this wear location, specifically at the flat part of the shin has been studied for reliability by Ruder M. 2023. 

- Kobsar, D., Charlton, J. M., Tse, C. T. F., Esculier, J. F., Graffos, A., Krowchuk, N. M., Thatcher, D., & Hunt, M. A. (2020). Validity and reliability of wearable inertial sensors in healthy adult walking: a systematic review and meta-analysis. Journal of neuroengineering and rehabilitation, 17(1), 62. https://doi.org/10.1186/s12984-020-00685-3

- Kobsar D, Masood Z, Khan H, Khalil N, Kiwan MY, Ridd S, Tobis M. Wearable Inertial Sensors for Gait Analysis in Adults with Osteoarthritis—A Scoping Review. Sensors. 2020; 20(24):7143. https://doi.org/10.3390/s20247143

- Ruder, M. C., Masood, Z., & Kobsar, D. (2023). Reliability of waveforms and gait metrics from multiple outdoor wearable inertial sensors collections in adults with knee osteoarthritis. Journal of biomechanics, 160, 111818. Advance online publication. https://doi.org/10.1016/j.jbiomech.2023.111818

Reviewer 2 Comments:

Thank you for your thorough review of our manuscript. Your insightful comments and questions have helped us refine our work. We have carefully addressed each of your concerns and we believe our manuscript is much improved as a result of these comments and revisions. The notes below highlight these changes and hope they make it easy to follow these revisions. 

Title 

1. I suggest to stress the fact that the population include older adults. 

Thank you, we have updated the title to specify that our study population consists of older adults. 

Introduction 

2. Please describe more in details the methods and findings of similar works focusing on the classification of physical activity levels (e.g., device locations, selected cut-points, methods to select them, etc.). 

These details have been elaborated on from lines 79 to 84, where this further discussion on how cut-points are determined from other studies. A study that I believe is a prime example of how physical activity is classified with accelerometers is by AHK Montoye et. al., (2020), where it aimed to develop cut-points from a wrist-worn ActiGraph in free-living adults, by comparing the activity count distributions for different activity types of known intensities. 

- Montoye AHK, Clevenger KA, Pfeiffer KA, Nelson MB, Bock JM, Imboden MT, et al. Development of cut-points for determining activity intensity from a wrist-worn ActiGraph accelerometer in free-living adults. J Sports Sci. 2020;38: 2569–2578. doi:10.1080/02640414.2020.1794244 

3. Please briefly define the activity characteristics mainly related to sedentary behavior, light, moderate and vigorous physical activity levels. 

Thank you, for this comment. Examples of these varying activity levels have now been provided from lines 72-76 within the introduction section. 

Methods 

Procedures 

4. Please cite a reference paper dealing with the choice of attaching the Actigraph above the right iliac crest (and not at L5 for instance) and explain the main reasons. 

This is a great point you make. The choice of wear location was clarified in lines 146-150. Essentially, this wear location comes from the Freedson et. al. (1998) paper which studies Actigraphy when the device is placed above the right hip. While the right hip is a general location, for consistency I instructed just above the right iliac crest for each participant. The right iliac crest is also an anatomical landmark which participants can find easily in case the ActiGraph device shifts during activities. This wear location for the ActiGraph has also been used by numerous other studies that also utilize the Freedson cut-points, which include AHK Montoye et. al., (2020) and AM Leinonen et. al., (2017). 

- Montoye AHK, Clevenger KA, Pfeiffer KA, Nelson MB, Bock JM, Imboden MT, et al. Development of cut-points for determining activity intensity from a wrist-worn ActiGraph accelerometer in free-living adults. J Sports Sci. 2020;38: 2569–2578. doi:10.1080/02640414.2020.1794244

- Leinonen, A. M., Ahola, R., Kulmala, J., Hakonen, H., Vähä-Ypyä, H., Herzig, K. H., Auvinen, J., Keinänen-Kiukaanniemi, S., Sievänen, H., Tammelin, T. H., Korpelainen, R., & Jämsä, T. (2017). Measuring Physical Activity in Free-Living Conditions-Comparison of Three Accelerometry-Based Methods. Frontiers in physiology, 7, 681. https://doi.org/10.3389/fphys.2016.00681

Data Analysis 

5. I suggest to explicitly mention the Freedson cut-points values. 

Thank you, these have now been stated on lines 181-182 and have been cited as well. 

6. The authors sequentially optimized the cut-points from the sedentary be

---

## [Decision Letter · Decision Letter 1]

22 Jan 2024

PONE-D-23-25684R1Validation of physical activity levels from shank-placed Axivity AX6 accelerometers in older adultsPLOS ONE

Dear Dr. Gafoor,

Thank you for submitting your manuscript to PLOS ONE. After careful consideration, we feel that it has merit but does not fully meet PLOS ONE’s publication criteria as it currently stands. Therefore, we invite you to submit a revised version of the manuscript that addresses the points raised during the review process.

Authors provided a detailed revision of the manuscript and it was constantly updated. However, some concerns that deserves to be clarified were pointed out bu the Expert. 

We look forward to receiving your revised manuscript.

Kind regards,

Andrea Tigrini, Ph.D.

Academic Editor

PLOS ONE

Journal Requirements:

Additional Editor Comments:

Authors provided a detailed revision of the manuscript and it was constantly updated. However, some concerns that deserves to be clarified were pointed out bu the Expert.

Reviewers' comments:

Reviewer's Responses to Questions

**Comments to the Author**

1. If the authors have adequately addressed your comments raised in a previous round of review and you feel that this manuscript is now acceptable for publication, you may indicate that here to bypass the “Comments to the Author” section, enter your conflict of interest statement in the “Confidential to Editor” section, and submit your "Accept" recommendation.

Reviewer #1: All comments have been addressed

Reviewer #2: (No Response)

2. Is the manuscript technically sound, and do the data support the conclusions?

Reviewer #1: Yes

Reviewer #2: Yes

3. Has the statistical analysis been performed appropriately and rigorously? 

Reviewer #1: Yes

Reviewer #2: Yes

4. Have the authors made all data underlying the findings in their manuscript fully available?

Reviewer #1: No

Reviewer #2: Yes

5. Is the manuscript presented in an intelligible fashion and written in standard English?

Reviewer #1: Yes

Reviewer #2: Yes

6. Review Comments to the Author

Reviewer #1: All the comments have been addressed.

I would suggest just to add a table summarizing the acronyms of the different PAs, some examples of them (as already written in the text) and the corresponding cut-points found by the author. Then, also to add the obtained cut-points in table 5 (or to create another table) near the values from Rhudy et al. just for a more direct comparison between what is found and what is present in the literature.

Reviewer #2: I would like to thank the authors for addressing the raised points. Further points below:

1) Concerning the previous raised point 7, it is clear how you performed the LOOCV analysis. But did you split the dataset into a construction set (to train and perform validation, e.g., 80% of the entire dataset) and a test set (e.g., 20% of the entire dataset)? You should show the averaged results from all the iterations of the LOOCV and also the results of the best trained model on the test set. Please update accordingly Figure 2, Table 2 and 3, and further explain how you evaluated the method performances. If you did not split the entire dataset as explained above, justify it and mention it in the limitations, and clarify what you mean by results from training model and LOOCV analysis.

2) Concerning the previous raised point 12, please mention the down-sampling at 30 Hz of Axivity data to be able to compare the results with ActiGraph.

3) The information of the percentual time spent in different PA intensities is provided by the Bland-Altman plots. Please add in the text the overall averaged reference % of time spent in different PA intensities to provide a clear idea of the different amount of data of the different PA levels. This values can be added in line 327.

4) Please rephrase lines 326-330. Since the activity counts are only based on accelerations, I would not say that the higher VPA cut-point at the shank is related to a higher effort of older adults, but only intensity.

7. PLOS authors have the option to publish the peer review history of their article (what does this mean?). If published, this will include your full peer review and any attached files.

Reviewer #1: No

Reviewer #2: No

---

## [Author Response · Author response to Decision Letter 1]

28 Feb 2024

Response to Reviewers 

February 28, 2024 

Reviewer #1: 

Thank you for taking the time to review our paper. We truly appreciate the comments made and have addressed each one below. In doing so, we again believe this has further improved the quality of our paper. 

1. I would suggest just to add a table summarizing the acronyms of the different PAs, some examples of them (as already written in the text) and the corresponding cut-points found by the author. 

Thank you for this comment. We agree that this will be an important table to summarize the information from the study for readings. This information has been added to Table 4, which previously only communicated the resultant cut-points. Now, this table reflects the intensity in full, which also displays the abbreviations in the parentheses, and displays the examples of activity for each intensity.

2. Then, also to add the obtained cut-points in table 5 (or to create another table) near the values from Rhudy et al. just for a more direct comparison between what is found and what is present in the literature. 

Thank you for highlighting this aspect of the table as it needed further clarification. The table was presented with data from Rhudy for the foot, ankle, and hip, but that study did not have separate data for the shank. Therefore, the shank values in Table 5 are those from the current study. While the text has been updated to clarify this in lines 344-345, it is not overtly clear in the table itself. Therefore, we have adjusted the table caption to clarify this and added information in each line to further highlight this. 

Reviewer #2: I would like to thank the authors for addressing the raised points. Further points below: 

Thank you for taking the time to review our paper. Your comments and suggestions have helped enhance the overall quality of the manuscript. Below you will find all comments addressed and incorporated in the updated paper. 

1. Concerning the previous raised point 7, it is clear how you performed the LOOCV analysis. But did you split the dataset into a construction set (to train and perform validation, e.g., 80% of the entire dataset) and a test set (e.g., 20% of the entire dataset)? You should show the averaged results from all the iterations of the LOOCV and also the results of the best trained model on the test set. Please update accordingly Figure 2, Table 2 and 3, and further explain how you evaluated the method performances. 

If you did not split the entire dataset as explained above, justify it and mention it in the limitations, and clarify what you mean by results from training model and LOOCV analysis. 

Thank you for this comment. It highlights that the explanations regarding the training set can be clearer. Additionally, we believe the previous explanation and edits to the methods section may have actually further convoluted the process by overly relying on the association to machine learning. Therefore, we reworked this entire section to help clarify this process entirely. Briefly, it is important to note that a conventional LOOCV in a machine learning model was not used here, but rather we computed cut-points in a manner analogous to this process. In doing so, all accuracy results are fully blind to that individual, and as such the results presented can be taken as a true validation of the model, with no data bleeding from training to validation/test. It is our hope that the new LOOCV section in lines 201-213 helps to more directly convey this message. 

2. Concerning the previous raised point 12, please mention the down-sampling at 30 Hz of Axivity data to be able to compare the results with ActiGraph. 

Thank you for suggesting this. The down-sampling at 30 Hz of the Axivity data has now been mentioned in lines 178-179. 

3. The information of the percentual time spent in different PA intensities is provided by the Bland-Altman plots. Please add in the text the overall averaged reference % of time spent in different PA intensities to provide a clear idea of the different amount of data of the different PA levels. These values can be added in line 327. 

These values have now been added to the manuscript to give readers more of an insight of how much activity was completed. On average, participants spent 79.70%, 18.60%, 1.46%, and 0.24% in SB, LPA, MPA, and VPA, respectively. These values were added from lines 315-316, as this information is referenced earlier in the discussion. 

4. Please rephrase lines 326-330. Since the activity counts are only based on accelerations, I would not say that the higher VPA cut-point at the shank is related to a higher effort of older adults, but only intensity. 

Thank you for highlighting this oversight. This has been modified in the updated document to only include intensity.

---

## [Editor Report · Decision Letter 2]

1 Mar 2024

Validation of physical activity levels from shank-placed Axivity AX6 accelerometers in older adults

PONE-D-23-25684R2

Dear Dr. Gafoor,

We’re pleased to inform you that your manuscript has been judged scientifically suitable for publication and will be formally accepted for publication once it meets all outstanding technical requirements.

Kind regards,

Andrea Tigrini, Ph.D.

Academic Editor

PLOS ONE

Additional Editor Comments (optional):

All the comments were addressed. The paper can be published.
---

## [Editor Report · Acceptance letter]

30 Apr 2024

PONE-D-23-25684R2 

PLOS ONE

Dear Dr. Gafoor, 

I'm pleased to inform you that your manuscript has been deemed suitable for publication in PLOS ONE. Congratulations! Your manuscript is now being handed over to our production team.

Kind regards, 

on behalf of

Dr. Andrea Tigrini 

Academic Editor

PLOS ONE